# The late Holocene demise of a sublittoral oyster bed in the North Sea

**Lasse Sander**[1]*, **H. Christian Hass**[1†], **Rune Michaelis**[1,2], **Christopher Groß**[3], **Tanja Hausen**[3], **Bernadette Pogoda**[1,3]

**1** Alfred Wegener Institute, Helmholtz Centre for Polar and Marine Research, Wadden Sea Research Station, List/Sylt, Germany, **2** Lower Saxon Wadden Sea National Park Authority, Wilhelmshaven, Germany, **3** Alfred Wegener Institute, Helmholtz Centre for Polar and Marine Research, Shelf Sea System Ecology, Bremerhaven, Germany

† Deceased.
* lasse.sander@awi.de

**Data Availability Statement:** The data underlying this study is publicly available in the PANGAEA database at https://doi.pangaea.de/10.1594/PANGAEA.927199.

## Abstract

A fossil oyster bed (*Ostrea edulis*) was recently encountered offshore Helgoland (German Bight). Oysters are important filter feeders in marine environments and their habitat structure supports a large associated biodiversity. The European flat oyster *Ostrea edulis* has historically occurred in vast populations in the North Sea, but declined massively in the early 20th century. The ecological restoration of *Ostrea* habitats is a current focal point in the North Sea. To better understand the mechanisms that caused the local collapse of the oyster population, this study investigated the size structure, weight, and age of the shells, along with the spatial dimensions, seafloor properties, and environmental context of the oyster bed. The results show that the demise of the population occurred around 700 CE, ruling out excessive harvest as a driver of decline. Synchronicity of increased geomorphological activity of rivers and concurrent major land use changes in early medieval Europe suggest that increased sedimentation was a viable stressor that reduced the performance of the oysters. The shells provided no indication of a demographically poor state of the oyster bed prior to its demise, but manifested evidence of the wide-spread occurrence of the boring sponge *Cliona* sp. Our study challenges the assumption of a stable preindustrial state of the European flat oyster in the North Sea, and we conclude that the long-term variability of environmental conditions needs to be addressed to benchmark success criteria for the restoration of *O. edulis*.

## 1. Introduction

The European flat oyster (*Ostrea edulis*) is a marine filter-feeding bivalve native to the larger North Sea area and historically occurred in vast populations occupying sublittoral environments across different habitats [Fig 1A; 1–4]. *Ostrea edulis* beds are characterized by a high structural complexity and provide shelter and settling ground for a range of associated species, thus increasing local biodiversity [5–7]. Oysters further hold both cultural and commercial value, and provide a range of ecosystem services positively affecting the overall quality of

**Funding:** This study received funding from the Federal Agency for Nature Conservation (BfN; www.bfn.de), the AWI Section "Coastal Ecology", and the project RESTORE, in the form of a grant (FKZ 3516892001) awarded to BP. We acknowledge support by the Open Access Publication Funds of Alfred-Wegener-Institut Helmholtz-Zentrum für Polar- und Meeresforschung.

**Competing interests:** The authors have declared that no competing interests exist.

marine and brackish environments [3, 8, 9]. Unfortunately, oyster populations have declined throughout Europe in recent historical times, which is attributed to the massive extraction and disturbance by targeted and non-targeted fisheries from the beginning of the industrial period [10–13]. In the mid-20th century, *O. edulis* became functionally extinct in the southern North Sea (i.e. >99% lost) [14–16]. Several national and international projects are currently working towards the reestablishment and restocking of *O. edulis* in European waters [9]. Small numbers of flat oysters are occasionally found in the North Sea, suggesting a restorative potential of the species in the area. However, it remains unclear, which environmental parameters have supported the development of a large population of *O. edulis* in offshore environments in the past [8, 17–19].

Despite the current interest in the restoration of *Ostrea* habitats, knowledge concerning the past distribution and ecological state of oyster stocks remains limited and largely conceptual [1]. Historical harvest statistics document the massive extraction of marketable specimen, and past abundance is often taken as proxy for the healthy state of oyster populations, a view that is further supported by the wide spatial distribution of *O. edulis* as indicated in historical maps [12–15, 20, 21].

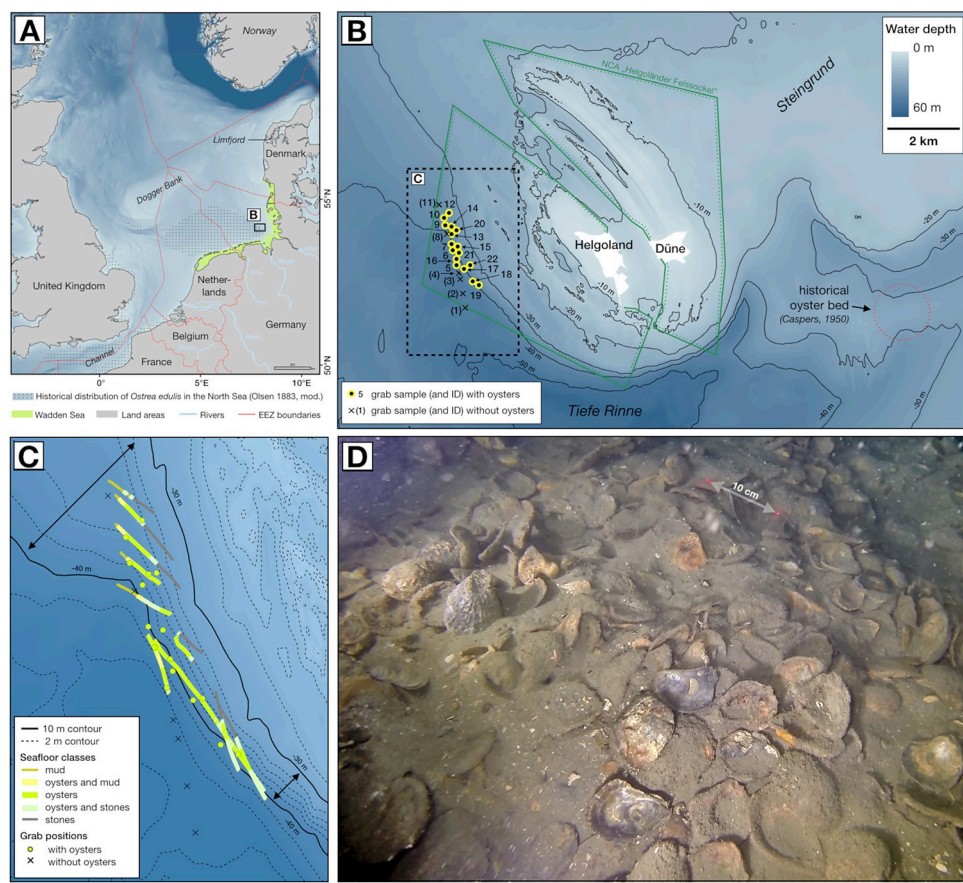

**Fig 1.** (A) Overview map of the North Sea basin with the historical distribution of *Ostrea edulis* [20], with background data from [42, 43], (B) Bathymetry and location of the studied oyster bed offshore Helgoland, with background data from [44], (C) Drift-video transects and the surface composition around the oyster bed, (D) Still-image example of oyster shells under a thin veneer of sediment (NB: laser for scale = 10 cm). See S1 Fig in S1 File for examples of all seafloor classes.

In the German Bight, a general distinction between the historical *Ostrea* beds of the Wadden Sea and the so-called offshore or deep-water populations (also referred to as "Helgoländer" [22]) has to be made. The environmental conditions in both areas are significantly different (e.g. water depth, hydrodynamics), and the intensity of exploitation varied largely. The amounts of oysters extracted from the Wadden Sea have been more than an order of magnitude higher than those harvested from the open North Sea and targeted near-shore fishery has been maintained considerably longer [15, 23]. In the late 18[th] century, large amounts of oysters were found in proximity to the offshore island of Helgoland but were not commonly fished due to the lack of adequate equipment [24, 25]. Oyster beds in the open North Sea probably remained unknown until the mid-19[th] century and their discovery was followed by an increase in the intensity of use [15, 26]. Until the 1870s, overexploitation had already decreased landings significantly and problems with recruitment had become evident [27]. In the year 1900, the oyster beds around Helgoland have been described as disproportionally old, characterized by the lack of juveniles and more than half of the population had been infested by the boring sponge *Cliona celata* [27]. Harvest was discontinued at the beginning of World War I, but the oysters did not recover [27].

Ecological restoration of the European flat oyster and its habitat is expected to result in the provision of a range of desirable ecosystem functions and services for the North Sea. These functions include an increased biodiversity and habitat complexity, which result in the provision of food, shelter, spawning ground and settlement substrate, along with the local enhancement of benthopelagic-coupling and water quality [28]. Accordingly, in the context of marine nature conservation, the ecological role of biogenic reefs and ecosystem engineers is valued superior. A number of trials are currently initiated in the southern North Sea [9, 16, 29]. The success is, however, contingent upon the existence of suitable habitat conditions that allow the development of a stable oyster population. Due to the substantial loss of oyster beds in the German Bight and the severely degraded state of the remaining populations, no profound ecological baseline and no reliable data on former biogenic reef structures in the sublittoral zone exist [13].

This study investigates a recently discovered fossil oyster bed located offshore Helgoland in order to facilitate a long missing description of a deep-sublittoral oyster bed in the North Sea. The aims of this study are (1) to obtain basic information on the location, composition, and age structure of the oyster bed, (2) to provide a new perspective on the past existence and state of sublittoral oyster habitats in the open North Sea, and (3) to discuss the long-term variability of potential environmental stressors (other than direct human impact) as a benchmark to inform current oyster restoration initiatives.

## 2. Methods

### 2.1 Study area

The object of this study is an oyster bed that was found 3.3 km west of Helgoland during a drift-video survey in summer 2017 [30]. The island of Helgoland is located in the German Bight (south-eastern North Sea) at a distance of approx. 40–45 km offshore from the East and North Frisian Wadden Sea (Fig 1A). Salinity in the waters around Helgoland ranges around 30–32 [31, 32] and the area is influenced by the discharge of freshwater from rivers (Ems, Weser, Elbe) [33, 34]. The south-eastern North Sea receives inflow of Atlantic waters, primarily through the English Channel, and currents from western and south-western directions dominate [35–38]. Tide-induced surface currents can reach velocities of up to >0.6 m/sec during spring tides [39]. The seafloor around Helgoland is gently inclined and diverse in its geological composition, including bedrock and rubble composed of Triassic sandstone,

Cretaceous limestone with flint, and Holocene sands and mud, which is also reflected in the bathymetry as well as in the presence of different bedrock, hard substrates and soft-sediment types [40, 41]. This situation determines the availability of a range of different habitat conditions for sessile organisms, such as water depth, current exposure, slope, and sediment composition, which define light penetration, phytoplankton concentration and food availability, as well as larval distribution. The fossil oyster bed is located entirely within the nature conservation area "Helgoländer Felssockel" (Fig 1B). A permit for research vessels to navigate the area was issued by the responsible Water and Shipping Authority Tönning in accordance with § 3 of the Ordinance on navigating in the nature reserve "Helgoländer Felssockel". The includes the permission to obtain samples from the seafloor.

## 2.2 Field data

**2.2.1 Drift video.** A ship-based survey of the fossil oyster bed was conducted on board RV "Mya II" between June 17–20, 2019. Video data was obtained using two ahead-oriented cameras (CT3009, C-Tecnics, Aberdeen, Scotland; resolution: 1920 x 1072, 30 frames per second; GoPro, HERO 3+ black edition, GoPro, Inc., San Mateo, California; resolution: 1920 x 1440, 47.95 frames per second) mounted to a metal frame. To improve ambient light conditions, four LED-lamps (CT4004, C-Tecnics, Aberdeen, Scotland) were attached to the frame along with two underwater lasers at a reference spacing of 10 cm. The video frame was deployed by a metal wire from the drifting ship (velocity < 1 knot) using a small crane. The camera frame was kept close to the seafloor and repeatedly lowered to the ground in order to improve the quality of the video material.

**2.2.2 Grab sampling.** Locations for grab sampling were selected based on the video data and chosen to represent areas of different oyster densities and seafloor compositions. Sampling was conducted using a HELCOM grab sampler with a volume of 40 liters. The contents of the grab were transferred into a plastic container. All oyster shells were picked and rinsed with seawater, including fractured specimen allowing an estimation of the original shell size and properties (i.e. preservation of the umbo and >50% of the estimated shell area). All other fragments were discarded.

**2.2.3 Background data.** The interpretation of the field data is supported by a digital bathymetric model for the German Bight [42] and additional data from AufMod [43] and GEBCO gridded bathymetric data [44]. Modelled current velocities for selected locations within the oyster bed were provided by the Operational Circulation Model (BSHcmod) [45].

## 2.3 Sample analysis

In the laboratory, the samples were dried at 48°C for 72 hours. All specimen from all samples were cleaned with a brush and weighed on a laboratory scale (in grams with an accuracy of two decimals). Samples were photographed under artificial light (Canon EOS 7D Mark II with Canon EF-S 18–55mm *f*/3.5–5.6 lens) from both sides (interior, exterior) and were measured (length, width) using ImageJ 1.52q [46]. The projected 2D oyster area was determined by highlighting the shell, using the "color threshold" tool (hue: 0–28, saturation: 0–243, brightness: 0–230, thresholding method: default, threshold color: red, color space: HSB), and a subsequent measurement of the oyster area by the "wand tracing" tool. To assess the number and area of boring sponge holes for each individual oyster, the original image was modified in color (converted to 8-bit, colorized with "spectrum", converted to RGB) and the identified holes were highlighted by color thresholding (see S2 Fig in S1 File).

## 2.4 Radiocarbon dating

Specimen of *O. edulis* from each grab location were selected for age determination based on size, and state of preservation. A total number of 19 samples were used for radiocarbon dating, composed of one large shell from each grab location and three additional juvenile shells from three locations selected based on spatial distribution. Material for dating was obtained from the outermost (youngest) part of each respective valve. An overview of all samples is given in Table 2. The sampling areas were mechanically cleaned from overgrowth and sampled with pliers. The material was dated at the MICADAS laboratory at AWI Bremerhaven and all samples were calibrated using OxCal 4.3 [47] and the Marine 13 calibration curve [48]. Due to the unknown temporal variability of the reservoir age in the southern North Sea, the global average of 400 years was assumed to apply and no additional corrections (ΔR) were used in the calibration process [49]. All ages are converted to the BCE/CE (Before the Common Era/Common Era) system to allow for easier reference with historical data and accounts from the literature.

# 3. Results

## 3.1 Location and surface properties

The fossil oyster bed is located in water depths between 33–44 m below NN (local ordinance datum) and has a maximum length of 3000 m and a width of 400 m. It is located at a marked break in slope determined by the transition from the bedrock base of the island of Helgoland to the deeper and relatively flat North Sea shelf (Fig 1B). The oyster bed is orientated along its length axis from NNW to SSE. The seafloor at the oyster bed is inclined at an angle of c. 1˚, while the slope is in similar water depths much less pronounced north of the oyster bed (c. 0.2˚), but remains steep south-east of the bed (Fig 1C). The encountered oyster shells thus form a slim band roughly aligned with the isobaths. Towards the lower (i.e. deeper) boundary of the bed, the oysters gradually dip under unconsolidated cohesive sediment, while the upper (i.e. shallower) boundary is characterized by a gradual transition towards hard substrates, with the presence of gravel- to boulder-sized material.

## 3.2 Composition of the oyster shells

All shells from the grab samples were disarticulated and none of the valves were aggregated. However, there was a large variation of shell morphologies represented, suggesting attachment to either shell material or hard substrates. The maximum shell length was 11.0 cm (mean: 5.5 cm), and the maximum width was 9.5 cm (mean: 4.6 cm; Fig 2). An estimated amount of >60% of the oysters reached maturity [50]. All size classes were present in the samples and both length and width show a bimodal distribution, with peaks in the length bins 3.0–3.5 cm and 6.5–7.0 cm, as well as in the width bins 2.0–2.5 cm and 6.5–7.0 cm. The weight of the individual shells was between <1 g to 125 g (mean: 19 g). About 34% of the samples weighed between >0 and 5 g, and 54% weighed between >5 and 35 g.

## 3.3 Boring epifauna

Boring epifauna were plentiful on the oyster shells, mostly from the boring sponge *Cliona* sp., which can be identified by borehole size and morphology [51]. Depending on the respective stations, only 10–35% of sampled shells were not infested (see Table 1), with no preferences for right or left valves. Infestation was evenly distributed over the complete shell area (see. Fig 3); in smaller size classes, boring was mostly located around the umbo. Number of holes per cm$^2$ and percentage of perforated area increased with size class and reach a maximum in oyster

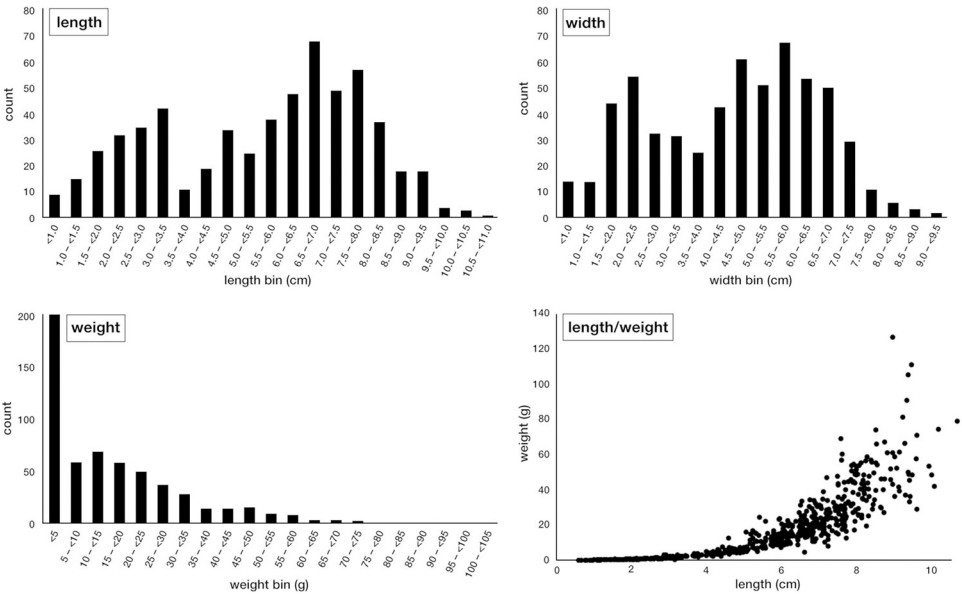

**Fig 2. Overview of the length, width and weight of all measured shells (n = 532) from the 16 grab sampling locations.**

shells with 9–10 cm length (see Fig 4). Some shells showed defense structures in the hypostracum layer evidencing that boring sponges impacted living oysters [51]. There was no relation between *Cliona* sp. prevalence and the depth or location of the grab samples.

## 3.4 Absolute age control

Almost all dated samples (n = 18) yielded calibrated radiocarbon ages between 2100 BCE– 700 CE (Table 2, Fig 5). One sample (a juvenile oyster; Sample 12J) yielded a median age of 220

**Table 1. Number and percentage of non-infested oyster shells per station: Inner and outer shell was assessed separately, as infestation was different (n = 1064).**

| Station | Total number of shells per station (infested and non-infested) | Number of non-infested shells | Percentage (%) of non-infested shells |
|---|---|---|---|
| 5 | 120 | 39 | 32.5 |
| 6 | 86 | 14 | 16.3 |
| 7 | 106 | 11 | 10.4 |
| 9 | 36 | 8 | 22.2 |
| 10 | 66 | 10 | 15.2 |
| 12 | 28 | 16 | 57.1 |
| 13 | 66 | 15 | 22.7 |
| 14 | 100 | 18 | 18.0 |
| 15 | 60 | 17 | 28.3 |
| 16 | 52 | 7 | 13.5 |
| 17 | 48 | 11 | 22.9 |
| 18 | 28 | 4 | 14.3 |
| 19 | 54 | 19 | 35.2 |
| 20 | 94 | 13 | 13.8 |
| 21 | 42 | 12 | 28.6 |
| 22 | 78 | 11 | 14.1 |
| | Total: 1064 | Total: 225 | Mean: 22.8 |

**Table 2. Overview of radiocarbon samples.**

| Sample ID | Lab ID | Sample bulk weight (mg) | Sample weight (µg C) | 14C age | Calibrated age range (2σ)* | % |
|---|---|---|---|---|---|---|
| 12J | 3027.1.1 | 82 | 1023 | 512±22 | 1700 CE– 1810 CE | 91.1 |
| 20/1-1 | 4399.1.1 | 272 | 979 | 1830±23 | 490 CE– 660 CE | 95.4 |
| 18J | 3031.1.1 | 38 | 1017 | 1900±23 | 430 CE– 580 CE | 95.4 |
| 19/1-1 | 4398.1.1 | 101 | 994 | 2194±23 | 90 CE– 240 CE | 95.4 |
| 2/1-6 | 4401.1.1 | 89 | 893 | 2369±23 | 140 BCE– 40 CE | 95.4 |
| 5/1-2 | 4389.1.2 | 41 | 963 | 2504±24 | 340 BCE– 140 BCE | 95.4 |
| 6/1-1 | 4390.1.2 | 121 | 966 | 2513±23 | 340 BCE– 160 BCE | 95.4 |
| 9/1-7 | 4392.1.1 | 104 | 988 | 2629±23 | 450 BCE– 300 BCE | 93.1 |
| 21/1-2 | 4400.1.1 | 126 | 921 | 2684±23 | 510 BCE– 360 BCE | 95.4 |
| 15A | 3028.1.1 | 77 | 1019 | 2698±23 | 530 BCE– 370 BCE | 95.4 |
| 15J | 3029.1.1 | 56 | 1020 | 2762±23 | 660 BCE– 410 BCE | 94.1 |
| 17/1-10 | 4397.1.1 | 378 | 933 | 2781±23 | 710 BCE– 450 BCE | 95.4 |
| 7/1-8 | 4391.1.1 | 134 | 994 | 2823±23 | 740 BCE– 530 BCE | 95.4 |
| 16/1-11 | 4396.1.1 | 134 | 736 | 2911±60 | 860 BCE– 530 BCE | 95.4 |
| 18A | 3030.1.1 | 132 | 983 | 2961±23 | 840 BCE– 740 BCE | 95.4 |
| 14/2-1 | 4395.1.1 | 107 | 983 | 3205±23 | 1170 BCE– 970 BCE | 95.4 |
| 12A | 3026.1.1 | 41 | 1035 | 3378±23 | 1380 BCE– 1210 BCE | 95.4 |
| 13/1-7 | 4394.1.1 | 235 | 908 | 3838±24 | 1930 BCE– 1730 BCE | 95.4 |
| 10/1-3 | 4393.1.1 | 273 | 748 | 3937±60 | 2160 BCE– 1800 BCE | 95.4 |

* in BCE/CE, rounded to the nearest 10, calibrated using the Marine13 calibration curve [48]

years (1800 CE). There is no apparent relationship between the depth or location of the sample with the determined radiocarbon age. From three locations, both an adult and a juvenile specimen were dated. Sample 12A and 12J, have very divergent ages with an offset of 3,500 years. For samples 18A and 18J, the age difference is at least 1170 years, while samples 15A and 15J have almost entirely overlapping age ranges (Table 2). Hence, there appears to be no relationship between the size (i.e. relative age) of the shells and their absolute age. The dated samples are interpreted as 19 independent replicas from an undisturbed and continuous seafloor feature and are therefore deemed to represent indication of the longevity of the oyster population at the investigated site. Thereby, the ages of the shells are interpreted as minimum ages for the existence of the bed and older shells are expected to exist at lower stratigraphic positions.

## 4. Discussion

### 4.1 The past ecological state of the oyster bed

The size structure of the material obtained suggests that the oyster bed was in a good reproductive state, given that all size classes are represented and that a large amount of specimen had apparently reached reproductive age (>60%) [50, 52–54]. The presence of small shells (down to spat size) suggests a renewal of the stock with young material from the own population or supplied from nearby, updrift locations. The presence of large specimens further indicates that enough settling substrate for spat and juvenile oysters had been present. Nevertheless, the investigated oyster bed does not represent a stratified record and no relative age inferences can be made between the specimen obtained in one grab sample. As the age of all shells has not been determined, the size and weight of the oyster shells is interpreted as a mixed record, presenting the time integrated product of the oyster bed's productivity over the period of its

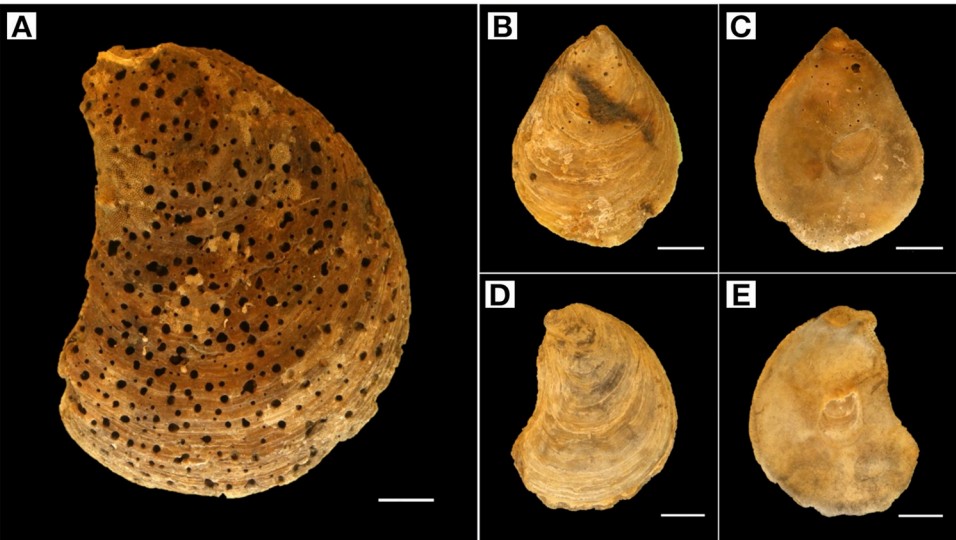

**Fig 3.** *Ostrea edulis* shells with boring holes of *Cliona* sp. showing different degrees of infestation: (A) adult shell, size class ≤9 cm with heavy infestation covering the complete shell, (B) and (C) outer and inner side of a smaller size class with infestation around the umbo, a typical sign for ante-mortem infection, (D) and (E) outer and inner side of an un-infested smaller size class. Scale = 1 cm.

existence. The inferences made below thus do not represent time slices, but reflect an overall assessment of the population structure and indication of its health.

Among other stressors, boring epifauna may have negatively affected the health of the oyster population. In the North Sea, the boring sponge *Cliona celata* infested living and non-living calcium carbonate substrates, such as limestone and bivalve shells of *O. edulis*, *Crassostrea gigas* and *Crepidula fornicata*. Living oysters are infested from a certain age and size, e.g. >3 years [55] as the sponge prefers rough and weathered crevices, which are scarce on younger

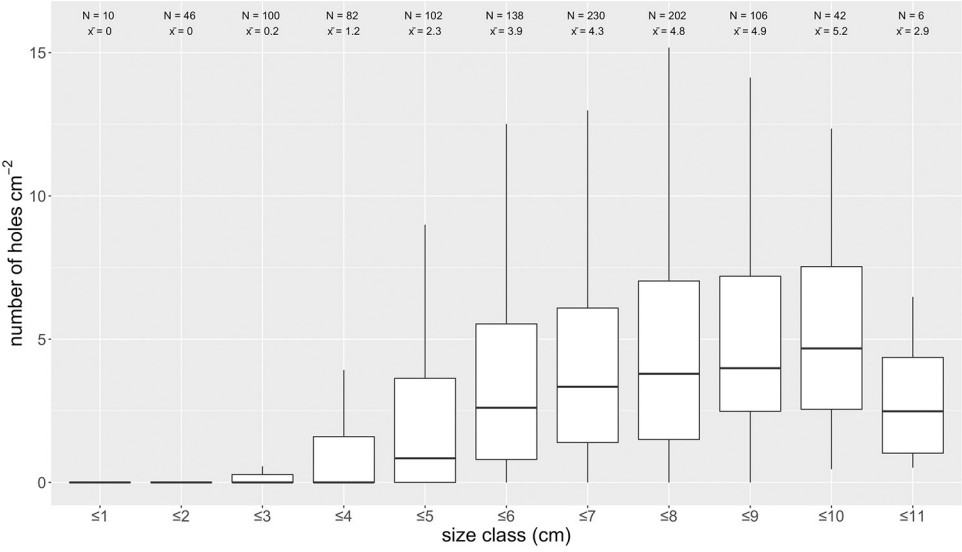

**Fig 4. Size classes of sampled *Ostrea edulis* shells and infestation with *Cliona* sp. distributed over different size classes (length) indicated as number of holes per area and individual.** Inner and outer shell infestation was assessed separately (n = 1064).

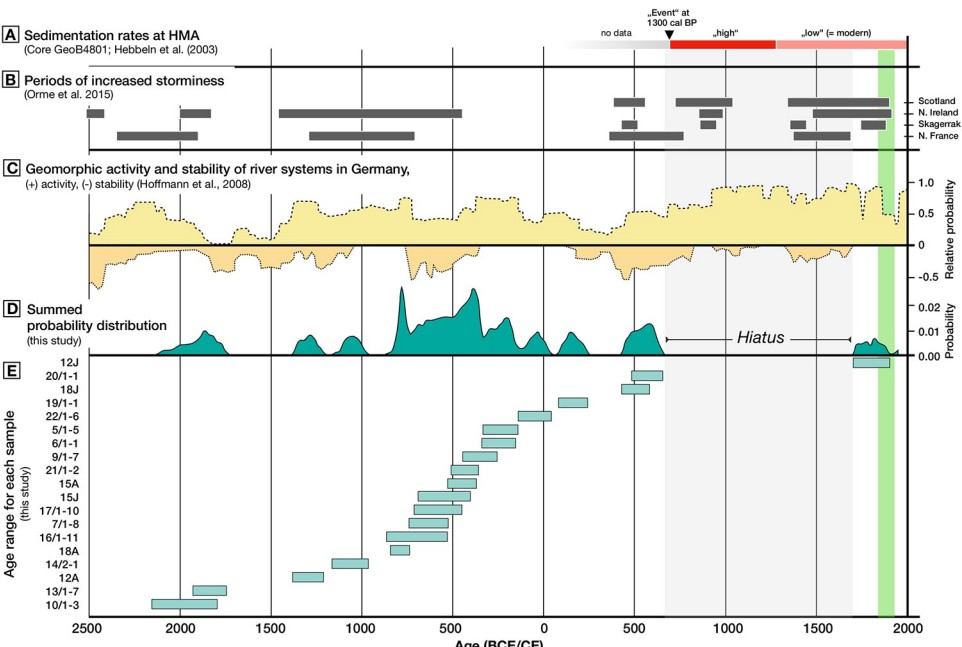

**Fig 5.** An overview of important environmental dynamics in the southern North Sea over the late Holocene (A-C) and the [14]C ages on shells from the studied oyster bed (D-E). (A) Changes in sedimentation rates in the Helgoland Mud Area (HMA) [65], (B) Periods of increased storminess from records in Atlantic Europe [73 and references therein], (C) A record of Holocene river activity (-) and stability (-) in Germany [69]. The green area to the right marks the time window from the first documentation of offshore oysters in the German Bight to their functional extinction [15].

smooth shell surfaces [51, 56]. The infection begins in the oldest part of the oyster around the umbo and protoconch [57]. In living oysters, the left valve is more affected by the sponge, eventually due to the infection by the larvae, which approach from the substrate below. Only when food availability and energy uptake are high during spring and summer, infested oysters can counteract the boring by building up hypostracum layers [55]. Postmortem, the complete shell is infested and boreholes are uniformly distributed [51]. If oysters have been infested by the boring sponge ante mortem, the growth and condition of the organism are reduced, which may result in elevated mortalities, especially under the critical presence of other stressors [58]. Several of the investigated oyster shells show evidence of secondary shell building mechanisms, thus indicating that an infestation took place ante mortem.

## 4.2 Causes of the late Holocene demise of the oyster bed

A number of 18 specimen yielded radiocarbon ages older than 700 CE, with the age of eleven sprecimen determined to calibrated ages between 900 BCE and 100 CE. Over the entire period from 1,400 BCE to 700 CE (2,100 years) the radiocarbon samples show an (almost) continuous record of overlapping age ranges, suggesting the sustained presence of oysters at the studied location. However, a hiatus in the oyster shell chronology is interpreted to indicate the demise of the population prior to 700 CE, probably entailing the collapse and disappearance of a living oyster bed at that particular location (Fig 5).

Long-term fluctuations in the health and density of oyster populations have been linked to sedimentation as an important environmental stressor for filter-feeding organisms [e.g. 8, 10, 14, 15, 59–61]. The data do not provide a lower limit for the establishment of the oyster bed, but appropriate habitat conditions have probably been established around Helgoland c. 8000

years ago, when rising postglacial sea-levels flooded the southern North Sea [62]. Radiocarbon ages from Limfjord support that *O. edulis* has been present in the North Sea area since about that time [63, 64]. It is thus hypothesized that older shell material at our study site may be obtained from lower (buried) beds, and that the absence of oysters older than 2000 BCE should rather be regarded as a sampling artefact than as an indication for the absence of oysters before that time (Fig 5). To our surprise, the age data suggest a demise of the oyster bank prior to the documented historical decline of North Sea oyster stocks, ruling out the destruction of the bed by overfishing as a cause. All (but one) dated oysters are older than ca. 700 CE. A single sample yielded a radiocarbon age falling into the time period of the known occurrence of oyster populations in the area. There is hence no evidence for the existence of an oyster bed between after ca. 700 CE and the historical documentation (19[th] century).

A study of sediment cores from the so-called Helgoland Mud Area, located southeast of the island, provided evidence for very high sedimentation rates of >13 mm/yr between 350–1200 CE [65]. After 1200 CE, sedimentation decreased to 1.6–2.6 mm/yr (Fig 5A) [65]. Sedimentation rates in the mud area had previously been estimated to 1.6 mm/yr as a Holocene average [66, 67]. It thus appears that the period of high sedimentation presents a marked deviation from the Holocene average, while the inferred rates for the period 1200 CE until present are in a similar order of magnitude as the Holocene background value. A potential explanation for the occurrence of a period of increased sedimentation rates may be found in major land-use changes on the European mainland [68]. Modelled results on the timing and extent of deforestation in Europe show that the conversion of forest cover to meadows and arable land proceeded rapidly and was at an advanced state already in the middle of the first millennium CE [68]. The timing is further consistent with a period of increased river activity in Europe starting at around 450 CE [69]. Anthropogenic land-use change may have led to a more direct coupling of land areas with riverine environment [70] resulting in an increase in turbidity and sedimentation in the south-eastern North Sea from fluvial sources (Fig 5C). In addition, geoarchaeological evidence suggests that a series of extreme river flood events occurred in the Elbe River after 550 CE [71]. At present, Elbe river plume fronts commonly reach the area around Helgoland [72]. The intensified occurrence of storms may have additionally increased sedimentation in the North Sea (i.e. sediment input from storm-induced coastal erosion) in periods of thermal transition in the late Holocene (i.e. warm to cold/cold to warm periods) [73]. Based on the temporal fit between the hiatus in the oyster bed and the above-described conceptual relations, we argue that increased sedimentation may be a viable candidate for explaining the deterioration of habitat quality and the subsequent demise of oyster population in the North Sea. Increased sedimentation results in ineffective filtration with a decrease in the quality and quantity of food uptake and a reduced energy budget, which affects all size classes of a filter-feeding population, and furthermore, the associated filter-feeding fauna.

Similar dynamics have been proposed to explain the decline of the Eastern oyster (*Crassostrea virginica*) in estuaries on the east coast of the USA. In mesocosm studies, C. virginica showed reduced biodeposition and condition index, as well as increased shell growth and higher mortality rates when buried in sediment [61]. Data from the Hudson River indicate that the absence of oysters in the late Holocene can be explained by increased estuarine sedimentation reflecting the heightened influx of terrestrial sediment during that time [74]. The material originated from soil erosion, induced by land-cover change and its delivery has been associated with wetter climate and the possible effects of rising sea level. In Chesapeake Bay, oyster populations declined and reef complexes shrank nearly 200 years ago under the pressure of increased oyster extraction, but sedimentation has contributed to the decline as an additional stressor [59]. More recent experiments and restoration efforts in Chesapeake Bay have emphasized the interaction between oyster reef height and sedimentation by the successful

performance of *Crassostrea virginica* reefs [75–77]. Based on archaeological reconstructions from kitchen middens, it has been hypothesized that a decline in oyster (*O. edulis*) consumption in southern Scandinavia around 4000 BCE, may be attributed to a decline in oyster abundance caused by increased sedimentation from coastal erosion [78]. Even though these studies demonstrate the negative impact of increased sedimentation on the health of oyster beds, other possible causes for a population decline should not be categorically excluded, such as a lack of larval supply, climate and current-pattern feedbacks, diseases (e.g. virus infections), or massive predation.

### 4.3 Implications for restoration

Our data describe the environmental context of a long-standing sublittoral oyster bed. Seafloor characteristics (e.g. slope angle and aspect) define habitat conditions (e.g. exposure to currents, bed shear stress, sedimentation) for site-specific oyster distribution [75, 76]. The integration of these identified factors with oyster larvae behavior [79] and into suitability index models [80, 81] will inform site selection and define connectivity of restoration sites [77]. The data furthermore suggest that a good state of the historical oyster populations in the German Bight prior to the dawn of industrial fisheries in the late 19th/early 20th century may stand less clear [e.g. 4, 15, 23] and that ideal habitat conditions for the restoration of *O. edulis* cannot be inferred from the historical/early industrial state of the North Sea. Early anthropogenic stressors, such as increased sedimentation from terrestrial erosion may be a factor that has previously been underestimated. Spatial shifts in the ecological quality of the environment, which can occur on shorter or larger time scales, drive the extension and health of oyster populations with regard to sedimentation rates and food availability. This stresses the importance of holistic approaches to inform the restoration and conservation of marine habitats [78, 82]. More specifically, the site selection for restoration measures needs to consider the potential impact of sedimentation as crucial factors [19]. Studies should address environmental stressors and their variability, as these present additional parameters that may jeopardize restorative efforts beyond problems of pure oyster biology and ecology.

## 5. Conclusions

The study of a fossil oyster bed in the southern North Sea showed that the native oyster (*O. edulis*) successfully lived for at least 2,500 years at a location where no living specimen are found today. The dimension and extension of the fossil oyster bed suggest that self-sustaining recruitment was possible and that environmental parameters limited the extent of the bed. The demise, however, appears to have happened more than 1300 years ago and the absence of living oysters cannot be explained by intensive fisheries and overexploitation. We propose that the local population went through a phase of deteriorating habitat conditions, primarily controlled by intensified sedimentation, which ultimately lead to the collapse of the oyster bed. This case study challenges the general assumption that the natural habitats of the North Sea have been in a pristine or optimal state prior to the industrial period. Early anthropogenic land-use changes may have had a larger effect on the performance of benthic filter-feeders than previously assumed, and the past period of success of *O. edulis* in the North Sea may have happened under more favorable conditions. These results imply that habitat deterioration by sedimentation may impede ecosystem recovery and that robust assessments of past ecological states can provide relevant baseline information for ecological restoration. The restoration efforts of *O. edulis* and *Ostrea* habitats thus need to consider the long-term variability of environmental conditions and past ecological states in order to achieve lasting ecological benefits.

## Supporting information

**S1 File.**
(PDF)

## Acknowledgments

The captain and crew of the research vessels FK "Uthörn" and FS "Mya II" are thanked for their collaboration and commitment during the field work. Gesine Mollenhauer and Torben Gentz (MICADAS, AWI Bremerhaven) conducted the radiocarbon dating. Lennard Kling-forth supported the data acquisition (shells/ImageJ). The BSH is thanked for providing mod-elled current data. H. Christian Hass passed away before the submission of the final version of this manuscript. Lasse Sander (the corresponding author) accepts responsibility for the integrity and validity of the data collected and analyzed. The academic editor, Romuald N. Lipcius (VIMS), and one anonymous reviewer are thanked for their input and comments, that helped to improve the overall quality and clarity of this manuscript.

## Author Contributions

**Conceptualization:** Lasse Sander, H. Christian Hass, Rune Michaelis.

**Data curation:** Lasse Sander, Rune Michaelis, Christopher Groß, Tanja Hausen, Bernadette Pogoda.

**Formal analysis:** Lasse Sander, H. Christian Hass, Rune Michaelis, Christopher Groß, Tanja Hausen, Bernadette Pogoda.

**Funding acquisition:** Bernadette Pogoda.

**Investigation:** Lasse Sander, H. Christian Hass, Rune Michaelis, Christopher Groß, Tanja Hausen, Bernadette Pogoda.

**Project administration:** Lasse Sander, H. Christian Hass, Rune Michaelis.

**Resources:** Lasse Sander, Rune Michaelis.

**Supervision:** Bernadette Pogoda.

**Visualization:** Lasse Sander, Bernadette Pogoda.

**Writing – original draft:** Lasse Sander.

**Writing – review & editing:** Lasse Sander, H. Christian Hass, Rune Michaelis, Christopher Groß, Tanja Hausen, Bernadette Pogoda.

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
