## [Decision Letter · Decision Letter 0]

13 Aug 2020

PONE-D-20-09992

A late Holocene oyster bed in the sublittoral zone of the southern North Sea: Drivers of its preindustrial demise and implications for restoration

PLOS ONE

Dear Dr. Sander,

Thank you for submitting your manuscript to PLOS ONE. We received one thorough review of your manuscript (ms) and I have reviewed it as well. After careful consideration, we feel that it has merit but does not fully meet PLOS ONE’s publication criteria as it currently stands. Therefore, we invite you to submit a revised version of the manuscript that addresses the points raised during the review process.

A rebuttal letter that responds to each point raised by the academic editor and reviewer. You should upload this letter as a separate file labeled 'Response to Reviewers'.A marked-up copy of your manuscript that highlights changes made to the original version. You should upload this as a separate file labeled 'Revised Manuscript with Track Changes'.An unmarked version of your revised paper without tracked changes. You should upload this as a separate file labeled 'Manuscript'.

We look forward to receiving your revised manuscript.

Kind regards,

Romuald N. Lipcius, Ph.D.

Academic Editor

PLOS ONE

Journal requiremets;

2. In your Methods section, please provide additional information regarding the permits you obtained to collect samples for the present study. Please ensure you have included the full name of the authority that approved the field site access and, if no permits were required, a brief statement explaining why.

3. Thank you for including your competing interests statement; "NO"

4. We note that your data statement includes that 'No - some restrictions will apply'. Please could you clarify the nature of these restrictions, ie. If due to ethical or legal reasons.

We note that you have indicated that data from this study are available upon request. PLOS only allows data to be available upon request if there are legal or ethical restrictions on sharing data publicly. For more information on unacceptable data access restrictions, please see http://journals.plos.org/plosone/s/data-availability#loc-unacceptable-data-access-restrictions.

5. We note that [Figure(s) 1] in your submission contain [map/satellite] images which may be copyrighted. All PLOS content is published under the Creative Commons Attribution License (CC BY 4.0), which means that the manuscript, images, and Supporting Information files will be freely available online, and any third party is permitted to access, download, copy, distribute, and use these materials in any way, even commercially, with proper attribution. For these reasons, we cannot publish previously copyrighted maps or satellite images created using proprietary data, such as Google software (Google Maps, Street View, and Earth). For more information, see our copyright guidelines: http://journals.plos.org/plosone/s/licenses-and-copyright.

1.    You may seek permission from the original copyright holder of Figure(s) [1] to publish the content specifically under the CC BY 4.0 license. 

Additional Editor Comments:

In attached, edited manuscript.

Reviewers' comments:

Reviewer's Responses to Questions

**Comments to the Author**

1. Is the manuscript technically sound, and do the data support the conclusions?

Reviewer #1: Yes

2. Has the statistical analysis been performed appropriately and rigorously? 

Reviewer #1: Yes

3. Have the authors made all data underlying the findings in their manuscript fully available?

Reviewer #1: Yes

4. Is the manuscript presented in an intelligible fashion and written in standard English?

Reviewer #1: Yes

5. Review Comments to the Author

Reviewer #1: This is a well-written manuscript analysing the age of a fossil offshore Ostrea oyster bed in the North Sea. Ostrea edulis has gone locally extinct in most of the North Sea, so investigating this fossil oyster bed is of broad interest for conservation biology. The main finding is that the oyster shells are mostly between 4000 and 1000 years old, suggesting the demise of this oyster bed prior to industrialization. The analyses are solid and the conclusions well-thought out. All minor changes are added into the manuscript as tracked changes.

My biggest issue is that the oyster shells look like an accumulation of shells, and the mixed ages of shells - rather than stratified layers - are also a bit concerning. I would like to see some indication of why the authors think that this is an in situ oyster bed, rather than oyster shells that were transported there by currents. Maybe currents aren't strong enough but this should be discussed.

6. PLOS authors have the option to publish the peer review history of their article (what does this mean?). If published, this will include your full peer review and any attached files.

Reviewer #1: No

---

## [Author Response · Author response to Decision Letter 0]

20 Oct 2020

Please refer to the attached file called "Point-by-point.docx" for a typeset version (incl. one figure) of this response: 

Point-by-point response

(PONE-D-20-09992)

Dear reviewers,

Thank you very much for your constructive comments and edits, which we have considered thoroughly in the preparation of the revised version of the manuscript. Below you will find a point-by-point response to your comments. We agreed to many of your edits and remarks and hence omitted to specifically respond to a few minor or entirely technical remarks. All edits were made using track-changes and you will find them in the attached document entitled “Revised Manuscript with Track Changes”. Our responses are consecutively ordered by line number in the revised manuscript and a reviewer ID is indicated in parentheses for easier reference (RL=Romuald N. Lipcius, R1 = anon. reviewer).

On behalf of all authors,

Lasse Sander

l.1f (R1): I prefer the short title over the long one, and would abandon the long title altogether

Reply: The title has been shortened as suggested.

l.12 (RL): Oysters are ecosystem engineers and dominant species, not keystone species. A keystone species is one whose ecosystem effects are disproportionate to its biomass, whereas the effects of oysters are proportional to their biomass/abundance.

Reply: Thank you for raising that question! Based on our current knowledge, it is of course debatable whether or not the European oyster should be regarded as a keystone species for the North Sea ecosystem – or whether a different terminology is preferable. Very little is known as to the actual abundance of O. edulis in the open North Sea and indication on the historical distribution of the species is rather unspecific regarding the per area densities of specimen (e.g. Olsen, 1883). Conceptually, however, the vast presence of a filter feeding bivalve on the seafloor of the open North Sea, as suggested by the historical accounts (ibid.), would have a considerable effect the shelf sea ecosystem and, especially when considering the effect on benthic-pelagic coupling, the term may most certainly be a valid choice to consider (which is not uncommon in the literature, e.g. Rodriguez-Perez et al., 2019 in Marine Pollution Bulletin 138).

However, we noticed that the term “keystone species” is only mentioned in the abstract (probably an artefact of an earlier version of the manuscript text) and we have therefore changed the sentence to read: “The European flat oyster Ostrea edulis has historically occurred in vast populations in the North Sea, but declined massively…“. Please excuse the confusion.

l.34 (R1): It is native to most of Europe. Please clarify

Reply: Yes, you are absolutely righty. However, in the context of this manuscript, dealing with the central North Sea, we feel that it is sufficient to point out that O. edulis is native to the area in question. We wrote “in the larger North Sea area” to include areas of similar dynamics within the North Atlantic (e.g. the English Channel, as referred to in e.g. Seaman & Ruth 1997) and its marginal seas (e.g. the Kattegat, as referred to in Lewis et al. 2016). Since the North Sea is well within the native range of O. edulis, we feel that discussing the full extent of the species’ distribution would reduce clarity and focus on past ecological dynamics within the North Sea. We hope you can agree.

l.62 (R1): Is this the location called „oyster grounds“? You refer to it later in the text, but should mention and explain it in this paragraph. Also, are the oysters around Heligoland the same oyster beds referred to here? This paragraph needs to be clearer

Reply: No, or at least not specifically. This sentence primarily serves to point out that the knowledge of the offshore oyster population in the open North Sea overall is relatively recent (as opposed to the documented use of intertidal oyster stocks since at least Roman times). The following period of use probably included, but was not limited to, the so called “Oyster Grounds”. We agree that a place reference is slightly confusing here and removed it from the manuscript text (see also: reply below, l.83). The sentence now reads: “Oyster beds in the open North Sea probably remained unknown until the mid-19th century and their discovery was followed by a massive increase in the intensity of use”.

l.83 (R1): This refers to a specific location. Please explain in more detail, preferably earlier in the introduction

Reply: This sentence has been changed for clarity and now reads: “(2) to provide a new perspective on the past existence and state of sublittoral oyster habitats in the open North Sea”. It was not intentional to refer to a specific location here.

l.89 (R1): Looking online, it seems the english spelling is Heligoland, see for example wikipedia. Please change throughout the text.

Reply: Both spellings, Heligoland and Helgoland, are used in English-language texts and are correct. We prefer to use the Danish/German spelling Helgoland, as it is more widely used in the scientific publications referred to in this manuscript. This includes place names such as the Helgoland mud area (Hebbeln et al. 2003), the nature conservation area Helgoländer Felssockel, or the journal Helgoland Marine Research (Springer).

l.103 (R1): Maybe refer to Fig. 1A in the introduction when you explain the different historical oyster locations

Reply: A reference to the figure has been added at the beginning of the introduction. 

l.105 (R1): I suggest to change the colors to highlight the presence of oysters.

Reply: In an earlier version of this figure, the locations with oysters were indicated with more contrast in the choice of color. However, we figured over time that the spatial distribution of oyster shells would be more adequately represented by visually less discrete colors. The upper boundary of the bed is a transition from oyster shells and stones to areas entirely composed of bedrock and stones. In this case, we can with absolute certainty say that no oyster shells were present from a certain point in the drift-video transects. We hence chose a thinner line in a shade of grey to indicate the absence of oysters. The lower boundary, however, is a gradual transition from oyster shells to unconsolidated sediment. The limit here is less clearly defined as detection is limited by the size of the grab sampler (depth penetration) and the thickness of the sediment layer. The oyster bed may therefore continue further to the west as suggested in the map by Olsen (1883; see also Fig. 1A). We hope this adequately explains, why we chose not to highlight the presence of oysters more discrete colors. 

l.141 (R1): In Fig. 5, you show an age range for each oyster. How did you calculate it, and what exactly is this age range? In other words: how did you determine the uncertainty around your age estimates?

Reply: The radiocarbon (or 14C) age is based on the measured ratio between two isotopes of carbon in a sample. One isotope is stable, but the other one becomes unstable once the organism dies. The 14C age thus uses the half-life of the unstable carbon isotope as a clock and could in our case be calculated to a certainty of around ±30 years. This value varies slightly and is individually assessed for each sample (see Table 2, column 5). Since the isotopic composition of ocean and atmosphere is variable over time and the measured value has to be corrected against a calibration curve (called Marine13, see asterisk below Table 2 and in column 6). This procedure increases the uncertainty of the age indication. It is a commonly agreed standard to provide ages as calibrated 2-sigma probabilities, expressed in age ranges (the oldest and youngest limiting age). Providing a 2-sigma uncertainty barely means that the real age of a measured sample will in 95,4% of cases be within the provided age range. These values are displayed in Figure 5 and are further given in Table 2, column 7. We hope this answers your question.

l.152 (R1): But you do not specify what kind of historical data in the M&M section. Please explain or remove from this section

Reply: The term historical data is merely used to describe all historical information and accounts referred to in the introduction and the discussion (such as: “Until the 1870s, overexploitation had already decreased landings significantly…”, l.67f). To improve clarity, the sentence has been changed to: “…and easier reference with historical data and accounts from the literature”. We hope this presents an adequate correction. 

l.165ff (R1): This result is not discussed. If the authors think it is not relevant, it should be removed from the MS

Reply: You are right. We will upload these data separately to the PANGAEA database. 

l.207 Table 1 (R1): Add a row showing the sums of columns 2 and 3, and the mean % from column 4.

Reply: Good idea. The information has been added.

l.244 (R1): Given that no shells were found attached to each other, and that there appears to be no stratigraphic layering of oysters from different millennia – which I would have expected – could these oyster shells have been drifted to this location? Please discuss.

Reply: Thank you for this comment, and yes, there may in fact be a stratigraphic layering present in the oyster bed! The problem is that it was not possible to retrieve a stratigraphically undisturbed sample with the methods available to us, but we hope to find a solution for this in the near future. We observed shell morphology suggesting attachment, but did not, as you correctly point out, find attached specimen in clusters. The ability and capacity of O. edulis to form reefs and the process itself is not fully understood yet. But recent findings (Merk et al., accepted, Aquatic Conservations) show that young oysters formed aggregations not only triggered by individual shell growth, but also by specific epifauna: oyster clumps were held together by e.g. Lanice conchilega. Substantial dead O. edulis reef structures in the Black Sea also show the presence of Sabellaria taurica that could have had the same function (ibid.). Overall, the position of the oyster bed within the seafloor topography suggests that is located at in a favorable position that could quite likely have supported a thriving oyster population. This inference is based on the presence of adequate settling substrate at a suitable water depth located on gently-sloping terrain perpendicular to the main direction of currents (at velocities of rarely above 1m s-1). The position is similar to a historically living oyster bed described by Caspers (1950), which supports our notion. Furthermore, do the sampled shells show virtually no signs of transport (such as rounding). Considering the water depth and the hydrodynamics of this part of the open North Sea, it is unrealistic that the empty oyster shells have drifted to the sampling site and accumulated in the densities we found them in.

l.246 Table 2 (R1): Define what the error is for 14C age, and what the % is for.

Reply: Please refer to our reply above (l.141)

l.342 Caption Fig. 5 (R1): The labeling is unclear. I would label the summed probability distribution as D and the indiivdual inferred age graph as E. The green arrow can simply be explained in this figure caption

Reply: This has been changed as suggested and the captions have been adjusted accordingly.

l.342 Caption Fig. 5 (R1): How did you calculate the summed probability distribution This needs to be described in material and methods

Reply: Please find below a screenshot of the calibration for one of the samples, which we here use to further illustrate the rationale behind probability distributions in 14C dating. 

As explained in the reply above, each measured 14C age has been corrected using a calibration curve (in blue), resulting in an age range with a probability to provide the correct age with of 95,4% (horizontal bracket). The certainty for each age within that range is equal, but each age-determined sample can alternatively be expressed as a probability distribution (grey curve). This distribution results from the combination of the uncertainty of the determined 14C age (red curve: here ±60 years) and the uncertainty of the calibration curve. Some ages within the two-sigma range thus may seem more likely to be correct, but statistically they are not! The purpose of summing these probability distributions for samples from the same location is exclusively to visually express the overlap between age ranges (or the lack of overlap!) and not to reduce the error margins of each individual age determination.

Summing probabilities is a basic operation in OxCal (the calibration program, Bronk Ramsey 2009, Radiocarbon 51). We have pondered your comment thoroughly, but have decided that it is not beneficial to explain this in more detail in the manuscript text, mainly for reasons of brevity, balance, and readability (for the broad readership of PLOS ONE). We hope that the explanations above help to make the procedure clear. We are of course open for applying changes, if you should feel these are necessary.

l.344 Caption Fig. 5 (R1): What does activity (+) and stability (-) mean? What does the relative probability refer to?

Reply: This data is taken from a study by Hoffmann et al. (2008, Quaternary Science Reviews 27) who compiled and reviewed age-determined indication on river activity or stability taken from the stratigraphic record of fluvial and colluvial deposits within river systems in Germany based on ~400 radiocarbon samples. The indication used by the authors are different sedimentary facies pointing towards either activity (such as sandy floodplain deposits from peak runoff events) or stability (such as soil formation on river banks). Since the study is based on radiocarbon dating, the indication of the individual samples is expressed by the combination of probability distributions (see replies above), with the difference that the authors here assign a positive value to all samples indicating activity and a negative value to those indicating stability. This results in a “swinging” range of positive and negative values. Relative probability is merely used as the unit the values are expressed in and should as a relative value be read as periods of more/less activity and periods of more/less stability in river systems over time.

Data like these are rare, and few studies systematically address past landscape dynamics in river systems or (more specifically to our interest) past sedimentation to an open shelf sea from adjacent river systems. Even though the data are less straight-forward to interpret than we could wish for, this is the best available information to infer late Holocene river-borne sedimentation in the North Sea. We made sure to interpret the data cautiously and to not overstretch the indication in absolute terms. 

l.360 (RL): See additional ref for HSI models. That ref reviewed all published HSI models for oysters.

Reply: Thank you for this comment. We included some additional information and refer to the references you suggested.

l.376 (R1): How did you confirm that the oysters formed a self-sustaining population?

Reply: We did not confirm this, but made sure to use careful wording here. The sentence states that “the dimension and extension of the fossil oyster bed suggest that self-sustaining recruitment was possible”. Your primary indication is the size of the oyster bed. We do not know much detail about the past abundance of oysters for each time slice within the time-period of the bed’s existence, but the sheer number of shells preserved on the sea floor today provides indication for periods of high productivity. The historical accounts (e.g. Olsen 1883) suggest, that the studied oyster bed Helgoland was located down-current from a large area of seafloor occupied by oysters, thus increasing the probability of frequent spatfall. The main point to mention this in the conclusions section is that nothing indicates a limitation in the size of the oyster bed by reproductive issues of the population for large parts of the late Holocene.

l.402 (R1): Please change all latin species names to italics, especially in the reference section

Reply: Thank you very much for putting so much attention to detail! This has been adjusted as suggested.

---

## [Editor Report · Decision Letter 1]

29 Oct 2020

The late Holocene demise of a sublittoral oyster bed in the North Sea

PONE-D-20-09992R1

Dear Dr. Sander,

We’re pleased to inform you that your manuscript has been judged scientifically suitable for publication and will be formally accepted for publication once it meets all outstanding technical requirements.

Kind regards,

Romuald N. Lipcius, Ph.D.

Academic Editor

PLOS ONE

Additional Editor Comments (optional):

Thank you for your revision, which is now acceptable for publication. I found the work interesting, particularly since I work on eastern oyster restoration.
---

## [Editor Report · Acceptance letter]

4 Feb 2021

PONE-D-20-09992R1 

The late Holocene demise of a sublittoral oyster bed in the North Sea 

Dear Dr. Sander:

I'm pleased to inform you that your manuscript has been deemed suitable for publication in PLOS ONE. Congratulations! Your manuscript is now with our production department. 

Kind regards, 

on behalf of

Professor Romuald N. Lipcius 

Academic Editor

PLOS ONE